# Profile of Patients with Cardiovascular Diseases during the Pandemic in a Cardiology Clinic of a COVID-19 Support Hospital

**DOI:** 10.3390/healthcare10101887

**Published:** 2022-09-27

**Authors:** Andrei Cârstea, Adrian Mită, Mircea-Cătălin Forțofoiu, Irina Paula Doica, Doina Cârstea, Ileana-Diana Diaconu, Anca Abu Alhija Barău, Liviu Martin, Maria Monalisa Filip, Andreea Loredana Golli, Maria Forțofoiu

**Affiliations:** 1Doctoral School, University of Medicine and Pharmacy of Craiova, 200349 Craiova, Romania; 2Department of Cardiology, Clinical Municipal Hospital ”Philanthropy” of Craiova, 200143 Craiova, Romania; 3Department of Medical Semiology, Faculty of Medicine, University of Medicine and Pharmacy of Craiova, 200349 Craiova, Romania; 4Department of Internal Medicine 2—Gastroenterology Compartment, Clinical Municipal Hospital ”Philanthropy” of Craiova, 200143 Craiova, Romania; 5Department of Pediatric Pneumology, National Institute of Pneumology “Marius Nasta”, 050159 Bucharest, Romania; 6Targu Jiu Branch, Faculty of General Nursing, (AMG), University of “Titu Maiorescu” Bucharest, 040441 Bucharest, Romania; 7Department of Surgery, Emergency Clinical Hospital “Dr. Ștefan Odobleja” Craiova, 200349 Craiova, Romania; 8Department of Medical Semiology, Faculty of Midwife and Nursing, University of Medicine and Pharmacy of Craiova, 200349 Craiova, Romania; 9Department of Public Health and Management, Faculty of Medicine, University of Medicine and Pharmacy of Craiova, 200349 Craiova, Romania; 10Department of Public Health-Epidemiology, Clinical County Emergency Hospital of Craiova, 200642 Craiova, Romania; 11Department of Emergency Medicine, Faculty of Medicine, University of Medicine and Pharmacy of Craiova, 200349 Craiova, Romania; 12Department of Emergency Medicine, Clinical Municipal Hospital “Philanthropy” of Craiova, 200143 Craiova, Romania

**Keywords:** SARS-CoV-2, cardiovascular diseases, COVID-19 virus, pandemic infection

## Abstract

Background: During the pandemic, our hospital became a COVID support hospital and consequently the cardiology clinic had restricted activity; thus, it received only suspect and/or patients confirmed positive with the various COVID-19 strains that were associated with a chronic/flaring cardiovascular pathology. Methods: Two batches of patients admitted during a one-year period were compared in the cardiology clinic over two different periods of time: BATCH I (1 April 2019 to 31 March 2020), in a non-COVID context (BATCH I N-COV) and BATCH II (1 July 2020 to 30 June 2021) comprising patients that presented with respiratory infection of SARS-CoV-2 (BATCH II COV-2), associated with chronic and/or acute cardiovascular condition. To determine the profile of the patients admitted in our clinic, we observed the following parameters: age, type of cardiac condition, and admission mode (for the N-COV group). Results: The data obtained as absolute numbers and as percentages in relation to the total number of admissions were presented in separate tables and graphs for both of the studied groups. Conclusions: The SARS-CoV-2 pandemic, in its almost two years of evolution, has divided the medical world in two main categories: COVID and non-COVID. Admission of the patients with chronic, but non-COVID cardiac conditions, in our case, dropped to almost one-quarter when we compared the two absolute admission numbers: 1382 in the year prior to pandemic compared with only 356 in the pandemic year. We believe that the number of deaths due to SARS-CoV-2 infection was infinitely higher than the reported ones and uncountable, in as much as COVID-19 did not kill only the infected patients, but it has also yielded a very large number of collateral victims among chronic patients who had no contact with the disease, but were unable to be admitted and treated for chronic heart disease.

## 1. Introduction

The study proposed by the authors considers a national public health problem, rather than a problem with purely medical and scientific implications, since public health has major socio-economic and humanitarian implications. The SARS-CoV-2 pandemic came upon mankind in the same way as all natural (biological) disasters: suddenly, somehow out of thin air, lightning-fast, ruthless, with extremely high infectiveness, and causing multiple casualties among the population. Therefore, we support the idea that this pandemic has taken us by surprise. It caught the medical world, the social, political, economic, and spiritual world unprepared; moreover, it stunned the average man, regardless of age, sex, skin color, education and social condition, geographical area and, especially, comorbidities.

The pandemic divided the world into two main categories: patients with COVID, who received the full attention of the medical science world, and the non-COVID patients, which many health units have forgotten, and whose condition and evolution no longer interested local or governmental medical institutions. Unfortunately, these patients entered a “shadow cone” and their emergencies (surgery, implantation of stimulator, valvuloplasty) were repeatedly postponed, many times, until the natural solution unfortunately took place.

During the pandemic, Craiova Municipal Philanthropy Clinical Hospital (classified by the Ministry of Health as a fourth-degree hospital) has become a COVID support hospital and, consequently, the cardiology clinic (with 35 beds) has seen its activity restricted; thereby, it received only suspected and/or patients confirmed positive with the various COVID-19 strains that were associated with a chronic/acute cardiovascular pathology (Order no. 555/2020 on the approval of the Plan of Measures for preparation of hospitals in the context of the coronavirus epidemic COVID-19).

We bring to the attention of the medical world the data we obtained, through statistical comparison of the profile of the cardiac patients admitted in the year before the pandemic, and the profile of the cardiac patients in the following year (associated with COVID-19 infection).

## 2. Materials and Methods

The study was performed by comparing two batches of patients admitted during a one-year period to the Cardiology Clinic of the Craiova Municipal Philanthropy Clinical Hospital in different times. BATCH I counted the admitted patients between the period of April 2019 and March 2020 (except April and May 2020), in a non-COVID context (batch I N-COV). In BATCH II, the enrolled patients had SARS-CoV-2 respiratory infection (batch II COV-2) and were admitted to the clinic between July 2020 and June 2021. Both batches were associated with chronic and/or acute cardiac conditions.

BATCH I N-COV. The first study group comprised 1382 patients, divided into the following age groups: under 40 years old, 41–50, 51–60, 61–70, 71–80, and over 81 years old (regardless of sex). Depending on the age groups, we studied the type of admission by emergency (acute or chronic-flaring condition), or with an admission letter from a specialized (chronic) ambulatory department.

BATCH II COV-2. The second group comprised 356 patients with both cardiac and SARS-CoV-2 infection. All patients during the pandemic were admitted through the emergency unit (the emergency being respiratory failure caused by the COVID-19 respiratory infection and not the cardiac condition), so there was no admission letter. Therefore, there is no admission table as for BATCH I N-COV.

The objective of our study was to observe a series of parameters in order to determine the profile of cardiovascular patients admitted in our clinic, regardless of the period of study. The preset parameters were the age, the type of chronic cardiac condition, and the admission mode. From the many cardiovascular pathologies, in the present study, we focused on the profile of seven major groups: arterial hypertension (HTN), ischemic cardiomyopathy (ICM), valvular, rhythm disorders (RD)—atrial fibrillation (AFib), atrial flutter (AFL), pulmonary thromboembolism (PTE); surgical—prosthesis, PTCA, bypasses; pacemakers; atherosclerosis (ATS) or arteriosclerotic vascular disease (ASVD) with sequelae stroke. We also made a general comparison between the two periods of study regarding mortality. We must mention that the cardiac condition in the profile of patients admitted with SARS-CoV-2 infection was the primary cardiac condition that the patient suffered of or the one that, at the time of admission, dominated with acute or flaring symptoms. It is known that cardiac disorders, especially in elderly people, or in those with obesity [1] and/or diabetes mellitus, are often afflicted by two or three symptoms. Thus, in their chronic evolution, they can influence or aggravate each other.

This study has been performed in accordance with the provisions of the Helsinki Declaration from 1964 (The 18th World Medical Assembly), revised within the framework of The 29th World Medical Assembly from Tokyo in 1975. We have respected the WHO principles regarding patients’ rights and the Law on patients’ Rights 46/2003. Additionally, the study received approval of the Ethics Commission of the Municipal Philanthropy Clinical Hospital with no. 18798/19.10.2021.

The diagrams (graphs) that illustrate the evolution trends of the various evaluated parameters, as well as the statistical comparisons between them, were performed using the “Graph” tool of “Word” and “Excel” modules of the Microsoft Office Professional Plus 2016 software (Microsoft Corp., Redmond, Washington, USA) package and the program type “Add on” XLSTAT v5.1 Free trial version for “Excel” module, trial version (Addinsoft, Bucuresti, Romania).

## 3. Results

As a first result, shown in Figure 1, we compared the total number of admitted patients from both of our studied groups.

In Figure 2, for the N-COV batch, the distribution of patients by age group, expressed as percentages, is given in relation to the total number of patients admitted in the studied year. The majority (45.23%) of patients with heart diseases that required admission were in the age group 61–70, followed by the age group 71–80 (28%) and by the age group 51–60 years (10.50%). All three age groups account for more than three-quarters (83.73%) of the total number of patients admitted in the year before the pandemic. The third significant group contained patients over 80 years old (13.82%).

The heart conditions that led to the admission of patients from the N-COV batch to the cardiology clinic are presented in Figure 3, which shows the distribution of these conditions in relation to age groups.

We also established the order of the heart conditions for which patients were admitted to hospital by the number of admitted cases: ischemic cardiomyopathy (38.13%), arterial hypertension (23.30%), and atrial rhythm disorders (20.54%) (Figure 4).

In Figure 5, we plotted the patients by admission and age group (for the patients admitted as an emergency). In this way, we can easily notice that the admission due to emergency far exceeds half of all admitted patients (63.6%), out of which around 50% are in the age group 61–70 (51.3%), followed by the age groups 71–80 (25.25%) and 51–60 (13.65%). These percentages were calculated out of the total number of annual emergencies.

Figure 6 consists of a graph that shows the distribution of cardiac diseases by admission mode. The percentages were calculated out of the total number of emergency and chronic admissions, respectively. It is clear that ischemic cardiomyopathy dominates (41.3%), as it is by far the most frequent heart condition for which emergency are made, followed by the arterial hypertension (33.3%) and atrial rhythm disorders (22.3%).

It should be noted that PTE is a severe illness, that can only be admitted as emergency. During the pre-pandemic year there were only 4 cases (0.28%) of hospitalizations. The cases were so rare because the patients with severe forms of PTE were mostly redirected to Emergency County Hospital, given that our potential for care was exceeded.

For the second group studied, the COV-2 BATCH, in Table 1, we can analyze the distribution of patients by age group, expressed in absolute and percentage numbers, given in relation to the total number of patients admitted in the studied year.

The prevalence of heart diseases in BATCH II can be observed in Figure 7. From this chart, we can conclude that there was a significant change in the cardiac pathology during SARS-CoV-2 infection. 

More specifically, in Figure 8, the distribution of patients admitted with different cardiac pathologies by age group can be seen.

In Figure 9, we have compared the percentage of patients, by age group, from both of our studied batches with a view of highlighting the major differences.

## 4. Discussion

We begin by mentioning that the idea of the study arose primarily after an approximate parallel between the two groups of study. Through that, it was noticed that there was an enormous difference between the number of hospitalizations in the cardiology section during a normal year before COVID-19 (1382 patients) and during a year that belongs to “the new normal”(356 patients). It appears remarkable that the number of hospitalizations had decreased by three-quarters.

As mentioned before, all hospital admissions in our pandemic year were performed through the emergency system, but none of the hospital admissions were due to cardiovascular diseases, but instead to SARS-CoV-2 respiratory infection and acute respiratory failure [2].

In the pre-pandemic year, more than half of the patients that were usually admitted in a regular cardiology clinic were young and elderly adults (age groups 51–60 and 61–70 accounted for 55.73%). If we follow the parameter age also for batch II COV-2, the highest incidence on admission was for the over 80s age group, at almost half of the total of patients admitted in one year (49.23%). After expanding the age range (patients between 71 and 80 years old and over 80 years old), the percentage reached was over 75% (77.86%) [3]. This was because elderly and very elderly people, prior to the pandemic, were cared for by the geriatrics, neurology, or hospital-hostels for seniors and did not reach the cardiology clinics [4,5].

Consequently, the profile of heart disease has undergone drastic changes.

On the one hand, chronic ischemic cardiomyopathy (ICM) was the predominant reason for admission in the pre-pandemic year (by 38.13%), followed by arterial hypertension (HTN) and atrial rhythm disorders (RDs), mostly atrial fibrillation (AFib) [6]. In the main age group (61–70 years old), these three conditions in total accounted for the highest incidence (82% of the total admissions).

On the other hand, among the patients of BATCH II COV-2 admitted to a cardiology section by their cardiovascular comorbidities, the situation was as follows: ATS was predominant first (near half of the total of admissions −44.87%), HTN was in second place (31.2%), followed at a distance by ICM (9.26%). It is most likely that all had the same atherosclerotic etiology [7,8]. Other heart diseases, such as valvulopathies and RDs [5,9], had a significantly lower incidence (4.22% and 6.5%, respectively). Thus, we can summarize that our second group consisted mostly of geriatrics and especially neurological patients.

Particular attention should be paid to PTE.

In the BATCH I N-COV, the number of PTE patients admitted to our clinic was very low—4 (0,28%)—simply because we do not retain patients with severe cardiovascular disease or with a potential for aggravation. They usually are redirected “per primam” to a higher degree hospital equipped with interventional cardiology ward and a fully equipped ICU.

In BATCH II COV-2, keeping in mind that SARS-CoV-2 infection predisposes to hypercoagulability (this is the reason for the presence in protocols of the administration of fractional heparin anticoagulants on the first day of admission [10]), there were 11 PTE cases, which, in relation to the smaller number of patients in the batch, was a significantly higher percentage (3.1%).

Regarding PTE, it is necessary to underline three aspects.

First, while batch I patients presented PTE as the main reason for admission (all by emergency), for patients from batch II, PTE developed in the first week after admission to the clinic, as part of the SARS-CoV-2 clinical picture, even if they received the usual heparin anticoagulant doses. Hence, this is the reason why the forms developed under anticoagulation were mild and all of them had a favorable evolution.

Second, the age of the patients who developed PTE was between 41–70 years old (batch I) and 41–80 years old (batch II), which was somewhat similar; it is worth mentioning that, in both batches, people under 41 and the very elderly (almost and over 80 years old) were not included avoided.

Third, as somehow expected, all patients with PTE, from both groups, were overweight.

The admission to hospital of patients with chronic cardiac conditions decreased to almost one-quarter during the pandemic compared to the pre-pandemic period

In pandemic, all the admissions through the emergency care unit, in the cardiology clinic, were for respiratory failure due to SARS-CoV-2 infection; none were for cardiological conditions.

In the year before pandemic, 63.6% of patients were admitted to the cardiovascular clinic for cardiovascular diseases as emergency cases.

In batch I N-COV, the prevalence was higher in mature adults, while in batch II COV-2, the prevalence was higher in elderly and very elderly patients.

The medical field during the pandemic comprised proportionately more geriatric patients than those with cardiac pathology.

During a non-COVID year, the heart conditions that dominated the medical field were: chronic ischemic heart disease, followed by arterial hypertension and atrial rhythm disorders.

Regarding the batch II COV-2, the clinical picture consisted mostly of atherosclerotic cardiovascular disease, followed by arterial hypertension and ischemic cardiomyopathy.

Pulmonary thromboembolism, although rare in pre-pandemic admissions, was over 10 times more frequent in batch II COV-2.

Whether they are COVID or NON-COVID patients, every working citizen pays their mandatory contribution to the public health insurance and should therefore have, according to the Romanian Constitution, equal rights to medical access to healthcare in case of need.

SARS-CoV-2 infection, in addition to being a huge medical problem and a huge national public health issue, has caused problems with humanitarian repercussions all over the world.

After the analysis of the results of our study we can only agree with the concerns of cardiologists around the world. What has happened, during these almost two years of the pandemic, to the patients with chronic cardiovascular diseases? What happened to the patients with heart failure [11], irregular heart rate, valvular and ischemic cardiomyopathies, rhythm disorders, the sudden deaths, reanimated at the brink, or patients who required cardiovascular surgeries or those requiring constant surveillance and rigorous monitoring of the treatment scheme?

Apart from the fact that patients with cardiovascular diseases have a specific treatment that could not be interrupted, a special problem was also raised by cases requiring the administration of antibiotics. In those situations, we had to heed the interaction between antibiotic and cardiovascular medication. In addition, patients hospitalized in the intensive care service required increased attention due to antibiotic resistance; this was particularly relevant to elderly patients.

The prescription of antibiotics in patients with cardiovascular diseases, SARS-CoV-2 infection. and bacterial superinfection was supervised by the epidemiologist in collaboration with clinicians and laboratory doctors. This ensured protection against the development of resistance to antibiotics, avoiding unwanted interactions between antibiotics and cardiovascular medication (anticoagulants, antiarrhythmics, antiaggregants, antihypertensives, positive inotropes), administering an individualized treatment and avoiding the development of infections associated with the medical procedure [12].

There are studies that show that plant tinctures, such as Tragoponis pratensis folium and Myrtilli fructus, because of their polyphenolic compound content, have a broad-spectrum antibacterial effect against gram-positive and gram-negative bacteria and can be adjuvants to antibiotics through a synergistic mechanism. Therefore, their use in patients with cardiovascular diseases that are associated with SARS-CoV-2 infection and bacterial superinfection could be taken into account [13].

## 5. Conclusions

The SARS-CoV-2 pandemic has divided the medical world into two main categories: COVID and non-COVID.

The period of the pandemic brought profound changes in the health systems of the affected countries and our country did not avoid these changes.

As we showed in our study, the pandemic restricted the access of patients with cardiovascular diseases to adequate care in hospitals and specialized clinics, as these hospitals and clinics were transformed into support hospitals with the aim of increasing the hospitalization capacity of patients with SARS-CoV-2 infection with all that their right to these cares is guaranteed by the constitution.

In other words, the health system in our country, as with health systems in other European countries, was completely overwhelmed by the COVID-19 pandemic and was unable to provide adequate medical care to patients with cardiovascular diseases without SARS-CoV-2 infection who then paid the price of lack of access to specialized medical care over a long period of time.

## Figures and Tables

**Figure 1 healthcare-10-01887-f001:**
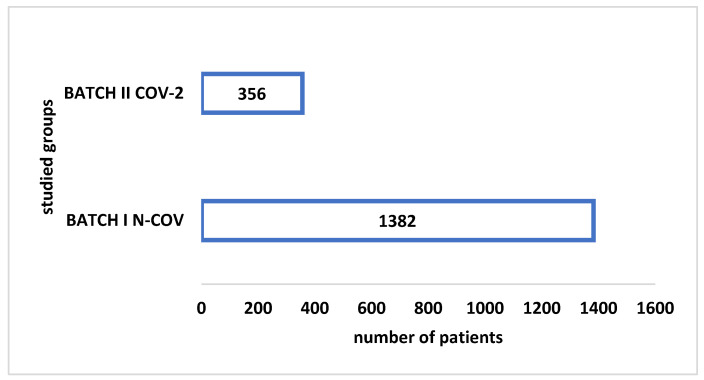
The number of admitted patients during pre-pandemic (BATCH I N-COV) and pandemic (BATCH II COV-2) years.

**Figure 2 healthcare-10-01887-f002:**
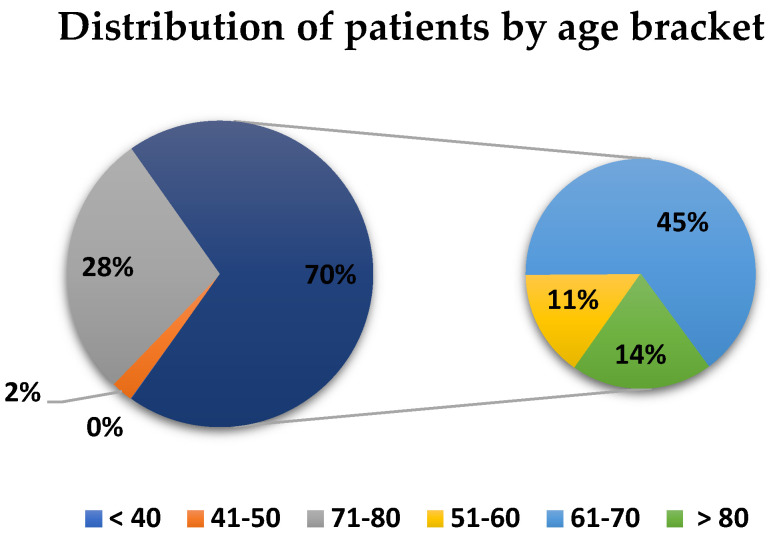
Distribution of N-COV batch I by age bracket.

**Figure 3 healthcare-10-01887-f003:**
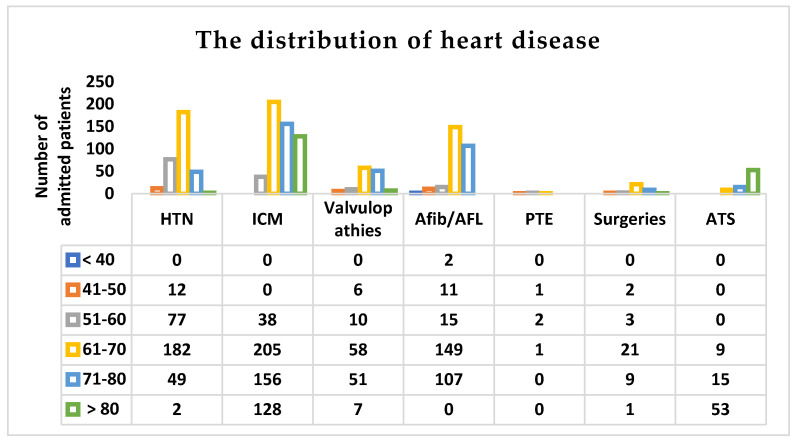
Graphical representation of the distribution of cardiovascular diseases compared to age groups for N-COV BATCH. HTN—arterial hypertension, ICM—ischemic cardiomyopathy, Afib—atrial fibrillation, AFL—atrial flutter, PTE—pulmonary thromboembolism, ATS—atherosclerosis.

**Figure 4 healthcare-10-01887-f004:**
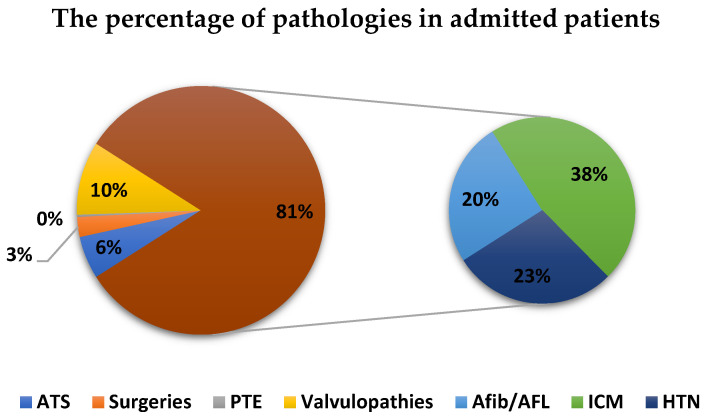
The distribution of heart disease in patients from N-COV BATCH. HTN—arterial hypertension, ICM—ischemic cardiomyopathy, Afib—atrial fibrillation, AFL—atrial flutter, PTE—pulmonary thromboembolism, ATS—atherosclerosis.

**Figure 5 healthcare-10-01887-f005:**
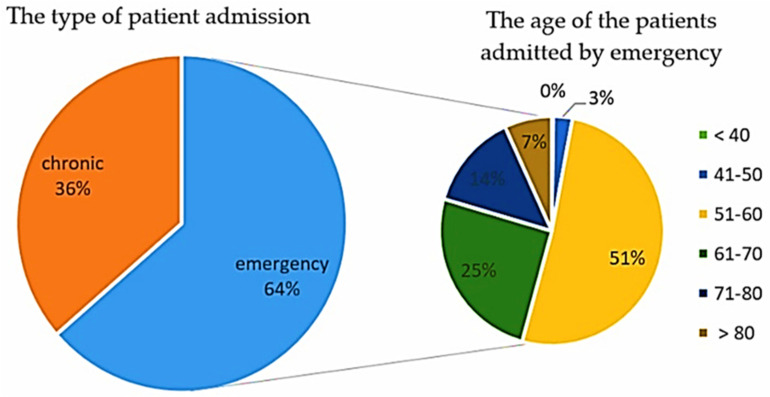
The distribution of patients by way of admission and age group (for emergencies) in N-COV BATCH.

**Figure 6 healthcare-10-01887-f006:**
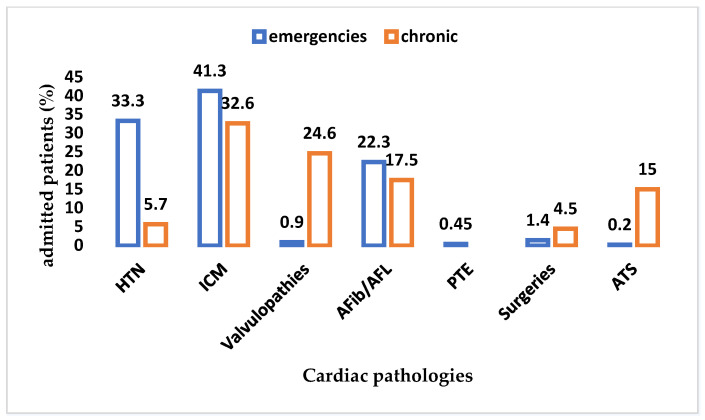
The distribution of cardiac disease by the admission mode in N-COV BATCH. HTN—arterial hypertension, ICM—ischemic cardiomyopathy, Afib—atrial fibrillation, AFL—atrial flutter, PTE—pulmonary thromboembolism, ATS—atherosclerosis.

**Figure 7 healthcare-10-01887-f007:**
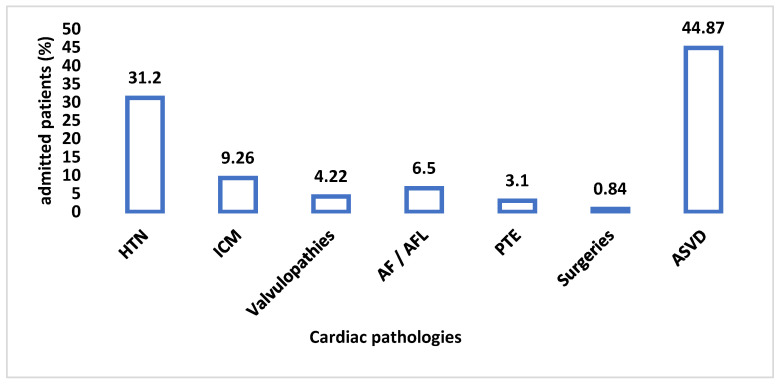
The prevalence of cardiac pathology in patients with SARS-CoV-2 infection. HTN—arterial hypertension, ICM—ischemic cardiomyopathy, Afib—atrial fibrillation, AFL—atrial flutter, PTE—pulmonary thromboembolism, ASVD—arteriosclerotic vascular disease.

**Figure 8 healthcare-10-01887-f008:**
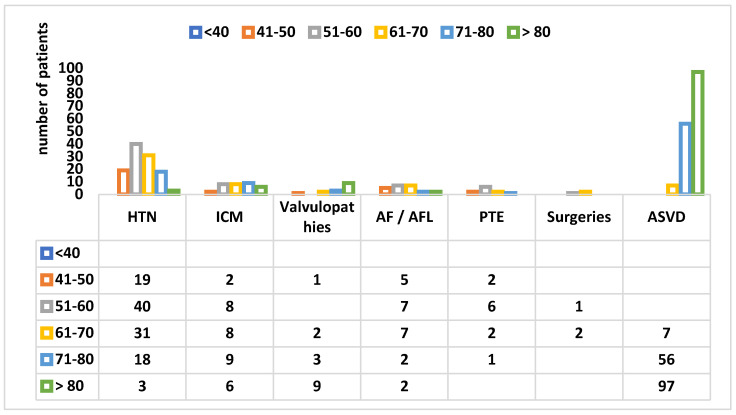
The distribution of cardiac diseases by age group in patients from COV-2 BATCH. HTN—arterial hypertension, ICM—ischemic cardiomyopathy, Afib—atrial fibrillation, AFL—atrial flutter, PTE—pulmonary thromboembolism, ASVD—arteriosclerotic vascular disease.

**Figure 9 healthcare-10-01887-f009:**
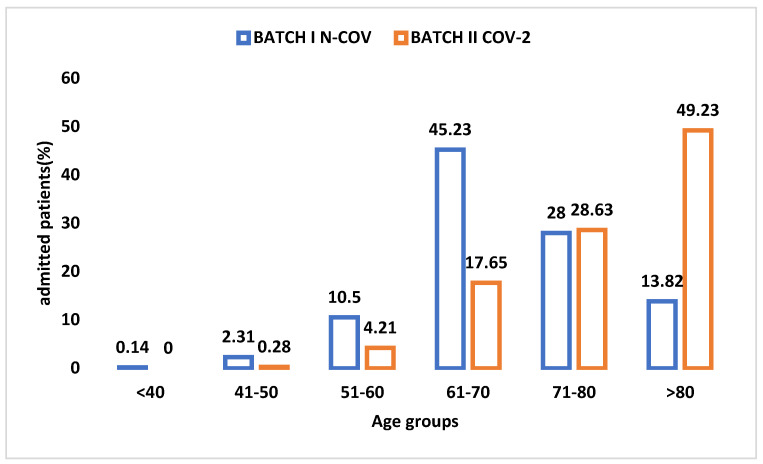
The distribution of patients by age group in both the studied groups.

**Table 1 healthcare-10-01887-t001:** Breakdown of batch COV-2 by age bracket.

Patient’s Age	<40	41–50	51–60	61–70	71–80	>80
**Number**	-	1	15	63	102	175
**%**	-	0.28	4.21	17.65	28.63	49.23

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
