# Peer review of "Profile of Patients with Cardiovascular Diseases during the Pandemic in a Cardiology Clinic of a COVID-19 Support Hospital"

_healthcare, 2022, doi:10.3390/healthcare10101887_

Round 1
Reviewer 1 Report
The article draws attention to a very important problem that causes similar consequences in all countries. Some corrective remarks:
· Please do not use exclamatory sentences in the Introduction, indicate the point to be emphasized in a different way.
· Abbreviations are missing from the Figures and some Tables, they must be added to the figure/table captions.
· Table 2 is very complicated and unclear, it is not clear that e.g. what quantities do the columns indicate. What does the top red line represent? Please make this table easier to follow.
· The age differences between the COVID and non-COVID patient groups should be shown in a separate figure/table, not separately.
· The significance levels of the differences are missing from each figure. The absolute values of the numbers are not relevant in this respect.
· The comparisons of the two groups (NCOV and COV) should be made so that the data of both groups can be seen in one figure.
Author Response
Hello,
Thank you for your time and effort in reviewing our article and for the suggestions made to correct it.
I have added the corrections you indicated to the article.
We hope that the corrected article will be better in terms of written text, figures and tables.
Thanks for the suggestions. We look forward to reviewing the corrected article.
Reviewer 2 Report
The article by Cârstea et al. studied two different types of population: patients with no covid (N-COV) and patients with SARS-COV-2 with regards to age, heart disease and admission mode, and discuss about the dire needs of care of non-covid patients at the times of pandemic. I have mentioned some of the changes needed in the article.
1. Figures 1, 2, 3 are very poor. The font in the graphs is not clear. Please change the fonts and label the x-axis and y-axis in the graphs.
2. Table 2 is confusing. Could the authors represent the data in other form?
3. Results and discussion sections are poorly written. Both sections need to be improved.
Line 161-162: Their absolute number, then presented as a 161 percentage, was compared to the total number of the group of affection and age group.
I am confused what the authors meant by group of affection
4. Please fix spelling and grammar errors. In many cases, I can see SARS-CO-2 or SARS COV 2 instead of SARS-COV-2. Please be consistent throughout the text.
Author Response
Hello,
Thank you for taking the time and effort to review our article.
We followed your advises and we performed the following changes:
- We improve the aspect and the manner of representation of the information, in order it to be better understood and easily followed.
- Table 2 was indeed hard to be comprehend, thus we discarded it and its information was expressed in other charts.
- In order to underline the differences between the studied groups, we added two more figures, one in the begining of the results and the other in the end.
- The information from table 4 from the previous article was represented currently in two different charts.
- The spelling and grammar errors were fixed.
- The abbreviations were added to the figures captions.
- The results and discussion sections were reviewed and improved.
We hope that the changes made are in accordance with your suggestions and that the current version of the article is much improved.
We look forward to the next review of our article.
Round 2
Reviewer 1 Report
The Conclusion should not contain detailed information, just summarize the results. Move them to the Discussion section.
Author Response
Hello,
Thank you for taking the time to review and for the suggestions made.
In accordance with your recommendations, I have moved those conclusions that express comments on the results to the Discussions section and reformulated the conclusions.
Reviewer 2 Report
The graphs look fine but it can still be modified to make it aesthetically pleasing/professional. For e.g. consistency with the fonts, modifying the y-axis and y-axis titles to make all the graphs consistent, removing the gridlines, etc.
Extensive English correction and spelling checks are required throughout the text. Some examples include line 162 which makes no sense at all, line 179 "observed" instead of "obeserved".
Author Response
Hello,
Thank you for reviewing this version of the article and for your suggestions.
I have corrected the graphics according to your recommendations. I hope that now they are more visible and clearer.